# Java Community Philosophy: More Children, Many Fortunes

Enung Hasanah 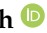

Department of Education Management, Universitas Ahmad Dahlan, Yogyakarta 55161, Indonesia;
enung.hasanah@mp.uad.ac.id

**Abstract:** In Indonesia, the island of Java is one of the largest and most populous regions. Indonesia's population was 271.35 million in 2021, of which 271.35 million or 55.19% live on the island of Java. Most families have more than two children because of the traditional philosophy in Javanese society that more children are linked to many fortunes. Many still believe in this philosophy, but others consider it an unsuitable inheritance of the colonial era. Therefore, this ethnographic study aimed to explore the development of the traditional philosophy of "more children, many fortunes" in modern Javanese society. The results showed that parents with more than two children from marginalized and wealthy families lived in cities and were highly educated. Several things support the eternal philosophy of "more children, many fortunes" in the life of the Javanese people. These include: (1) It is against God's decree to regulate births using contraception. (2) People believe that all children are born with their fortune. Therefore, parents should not worry about meeting the needs of many children. (3) Children are viewed as luck and eternal binders in domestic relationships. In this case, infertility is a potential source of family problems resulting in divorce. Therefore, many adopt children to avoid problems in household relationships.

**Keywords:** Javanese philosophy; culture; children; family; modern

## 1. Introduction

Indonesia is a multicultural country with ethnic, religious, and regional languages (Ikhsan and Giwangsa 2019), as well as ethnicities and 730 language groups spread over 17,744 islands (Agustina et al. 2019). Furthermore, Indonesia is one of the largest archipelagic countries in Asia, with 3.51% of the world's population (Worldometer 2022). Since the country has an area of 1.92 million square kilometers, the population density is 141 people per square kilometer.

Data from the Indonesian statistical center bureau indicated that the population density was still concentrated on the island of Java until 2021. However, the geographical area of the island is only about 7% of the country's entire territory. It is inhabited by 151.59 million people, equivalent to 56.10% of the population (BPS Indonesia 2021). Following this, the country will soon enjoy a demographic bonus because it has a large population of productive age (Sutikno 2020). On the contrary, the demographic dividend has not been accompanied by increased human resource quality due to Indonesians' low literacy ability (Mulya et al. 2018).

An imbalance in population density between Java and other islands also causes various economic and social problems. A small population in a large area is a potential cause of problems regarding poorly managed land (Nur Amrin et al. 2021). Meanwhile, overcrowded areas are vulnerable to more criminal acts when not balanced with improved quality of community life (Sabiq and Nurwati 2021). This indicates that the government needs to solve the problem of population inequality between regions to support the nation's development process.

The Indonesian government has initiated several programs to overcome the problem of population inequality. These include: (1) Transmigration program implemented in 1955, which is a program of moving people from densely to sparsely populated areas;

(2) Family planning program (KB), which aims to limit the number of children in a family and inhibit population growth. This is to be realized through the use of contraceptives to regulate the birth of babies in couples of childbearing age (Suryani and Suraili 2021). The family planning program began with the formation of the Family Planning Association on 23 December 1957 in the Indonesian Doctors Association building. Its name developed into the *Perkumpulan Keluarga Berencana Indonesia* (PKBI) or the Indonesian Planned Parenthood Association (IPPA). The PKBI fights for the realization of prosperous families through regulating or spacing pregnancies, treating infertility, and providing marriage advice. Based on the Presidential Instruction on 11 October 1968, Menkesra issued Decree No. 35/KPTS/Kesra/X/1968 concerning the formation of a team to prepare for establishing a Family Planning Institution. After holding the Menkesra meetings with other ministers and community leaders involved in family planning efforts, the National Family Planning Institute (LKBN), a semi-government institution, was formed on 17 October 1968 with Decree no. 36/KPTS/Kesra/X/1968.

The implementation of the family planning program has become Indonesia's development jargon since the 1980s. This is reinforced by Law Number 52 of 2009 concerning Population Development and Family Development (Presiden Indonesia 2009). The law defines family planning programs (KB) as an effort to regulate the birth of children. It also defines the ideal distance and age to give birth; and regulates pregnancy through promotion, protection, and assistance by reproductive rights to create a quality family. The program aims to meet the demand for family planning programs (KB) and reproductive health (KR). Another goal is to control the birth rate, improve the population's quality, and create quality small families (Dewi 2016).

Although the government has promoted family planning programs in various ways, the baby birth rate is still relatively high (William 2020). Most provinces still have high fertility rates due to low women's autonomy in making decisions on reproductive rights. Meanwhile, the binary logistic regression analysis showed that 7 of 11 independent female autonomy and other control variables significantly affected fertility. Furthermore, the independent variables affect fertility in Indonesia. This implies women's autonomy should be increased to support population control (Synthesa 2021). However, the government's efforts to evenly distribute the population have not been successful (Wardani and Arnellis 2019). In some cases, young families want more than two children, increasing the birth rate yearly.

The local community's culture influences the high birth rate regarding the value of children in the family (Laily et al. 2012). In North Sumatra, women with sons and no daughters are more likely to use modern contraceptives compared to women who have no sons yet. Sex composition also affects the use of contraceptives in Central Java because some parents want to have boys and girls (Trisnani 2020).

Javanese families have a community motto, "more children, many fortune (Muntamah et al. 2019). The philosophy of "more children, many fortunes" emerged in Javanese society during the Dutch colonial period. The regulation issued by Governor General Johannes van den Bosch in 1830 required each village to set aside 20% of its land for cultivating export commodities, especially tea, coffee, and cocoa. The crops would be sold to the colonial government at a fixed price. This condition is unique and exciting because the *cultuurstelsel* policy implemented by the Dutch colonial government on farming communities in the Madiun Residency had a high demographic impact. The high demographic figures were intentionally done to meet the large number of workers needed in agro–industrial plantations, especially sugar cane and coffee. This condition was behind the occurrence of the emergence of the "more children, many Fortune" philosophy in farming communities. Farmers want the tax burden imposed on them as labor to be divided by having many children (Izzah 2017). Many still believe in this philosophy, though others consider it an unsuitable inheritance of the colonial era. Therefore, this ethnographic study aimed to explore the development of the traditional philosophy of "more children, many fortunes" in modern Javanese society. It intended to investigate why parents have more than two children.

## 2. Context

This study was conducted in Yogyakarta, a densely populated city on the island of Java, Indonesia (Hasanah et al. 2019). All participants were parents from families living in a Javanese cultural environment, though some were residents that moved in from other cities on the island. The indigenous people of Yogyakarta hold Javanese cultural values and norms (Sari et al. 2019). Javanese society has a noble culture and conditions with the meaningful philosophy of life (Gaffara et al. 2021). Social norms and values become benchmarks in assessing a person's behavior and identity (Asnani et al. 2019). The first principle of Javanese culture is its hierarchical arrangement, where people have positions according to their levels. When speaking and behaving, Javanese people must adjust to the degree and work position of others.

Furthermore, the principle of respect for all relationships in society is arranged hierarchically; everyone is familiar with their place and their task. Those with higher positions must have fatherly or motherly traits to those whose position is lower (Hermawan et al. 2018). Javanese people believe life is a process of achieving inner perfection[1]. Society believes humans must curb their passions to achieve inner and outer perfection. Additionally, humans must be patient and not complain because they cannot solve life's problems. This is because patience is a part of Javanese behavior, including in family life.

The Javanese mindset does not define "what is life" but is more preoccupied with the issue of how humans reach *jumeneng* as '*teungguling* behavior' (*jumeneng* is a Javanese term that means the coronation of a king or queen, and *tetungguling* is a term in the Javanese language that means the King of the land of Java). The family behavior system has a cosmic philosophical meaning, a balance between the microcosm and the macrocosm. The balance implies the interrelationship of individual and social life between oneself and others (Wuri Arenggoasih 2021). It means that marriage is mandatory to perpetuate offspring in Javanese culture; the their parenting process it oriented towards raising children as ideal people. Therefore, parents are motivated to educate and raise their children accordingly, forming the priyayi group (Hasanah et al. 2019).

Although the Javanese communities think of raising their children into a priyayi, which takes time, resources, and effort. Moreover, in this modern society, the paradigm of having children has been shifted, even at governmental levels. Many families have more than two children. This research tries to find why modern Javanese families have many children.

## 3. Method

This study is ethnographic (Harrison 2020), a distinct research and writing tradition that, in this case, involved 15 participants identified through purposive sampling (Andrade 2021; Campbell et al. 2020). Samples were selected through various stages. First, the study polled parents of Javanese descent via Facebook about their opinion regarding the philosophy of "more children, many fortunes". In the second step, the study contacted people who commented on the philosophy on Facebook and asked them to participate. The participants' criteria are parents with more than two children of Javanese ethnicity who are willing to participate voluntarily in this study. The WA group link was sent to people that responded to questions on FB as potential participants. Two weeks later, 15 people had voluntarily agreed to participate in this study and were entered into the participant WA group. This study used 15 participants because it aimed to gain a deep understanding of the phenomenon, not to be generalized to a certain population (Gentles et al. 2015). The search for participants involved obtaining informed consent[2] and explaining the study process and objectives. Moreover, the consent process indicated the participants' rights and obligations in taking part in the study.

Data were collected through in-depth individual interviews (Hardavella et al. 2016) and field observations (Becker and Blanche 2020). The data were then analyzed using the thematic qualitative analysis method (Heydarian 2010) with the software ATLAS.ti 9 (Paulus et al. 2019). The analysis steps included: (1) entering transcripts and observation

data into the ATLAS.ti application; (2) coding; (3) developing themes based on coding results; (4) discussion and interpretation of data (Hasanah 2019; Hasanah et al. 2022).

## 4. Results

The results showed that along with globalization (Wittmann 2014), new ideas emerged regarding the ideal number of children for a prosperous modern family. Nevertheless, the philosophy of "more children, many fortunes" is still widespread in several community groups in Indonesia. This motto became a reference point in deter-mining the number of children in a modern Javanese family. The people believe in the philosophy of life, where many children are understood to attract luck in both the marginalized and the upper middle class.

The data analysis showed that the philosophy of "more children, many fortunes" still exists in the lives of some modern Javanese communities. Figure 1 describes the rea-sons families have many children.

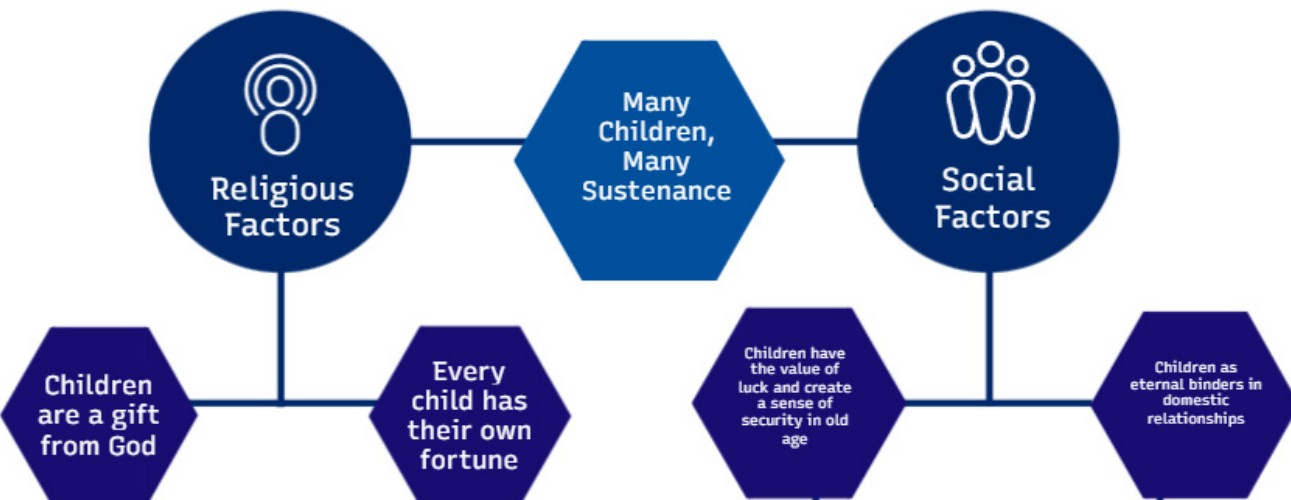

**Figure 1.** Factors influencing the decision to have many children in modern Javanese families.

Due to religious and social–cultural factors, some modern Javanese families have many children (Hasanah et al. 2022). Javanese society is religious and believes God is an almighty substance with power over human life (Kasnadi and Sutejo 2018). In contrast, the society is an Asian nation closely holding eastern culture (Memmi 2019; Ramesh et al. 2017) in their family life.

### 4.1. Children Are a Gift from God

Children are a gift from God, meaning regulating births using contraception goes against God's decrees. Many participants stated as such when asked why they did not regulate the number of births in their family. From a social–cultural perspective, this usually happens in the religious Javanese society (Wuri Arenggoasih 2021). They believe that everything happening in human life results from God's intervention, including the number of children in the family. In this study, 14 of the 15 participants stated that the God entrusts children because not all families are blessed with children. This implies that anyone entrusted by God to have children is not justified to reject them. Therefore, the participants did not want to use contraception (Boog and Cooper 2021) to regulate the number and spacing of births. This perception is seen in the relevant statements from the participants, such as the following statement from P1, a food entrepreneur in the Sleman area, Yogyakarta:

> Since the beginning of our marriage, we have agreed that we want to live well and build a household according to our religious values. This includes the number of children in the family. We do not use contraceptives because we believe it is

God's right to determine the number of children in the family. Since we do not want to violate God's decree, we do not follow the family planning program. My wife and I started a small business, and Alhamdulillah (praise to God), our company continues to grow with time. God also trusted us to have five children, three boys and two girls. Alhamdulillah, our lives are fulfilled even though we have many children. Yes, that's it, fortune has already been arranged, so I did not allow my wife to take family planning because it violated God's decree. Children are a blessing. When they get to be born, then let them be born. When we do not have a child, we cannot do anything. I do not want to refuse God's fortune to our family. (P1, lines 4–12)

Another participant, P8, a lecturer at a university, also stated as follows:

Yes, I have eight children, two girls and six boys. Alhamdulillah, healthy and *shalih shalihah* ("shalih and shalihah" is Islamic term for "benevolent"). Since I was young, I have never joined any family planning program. I will accept it with pleasure no matter how many children God gives. For me, limiting the number of children is going against the deed God has set. In our belief, preventing a child's birthright is a despicable act. (P8, lines 4–8)

P12 explained the reasons for having many children in the family:

I firmly believe that children by themselves are God's fortune for humans. Allah says, "Do not kill your children for fear of poverty; we will provide fortune for you and them." (Surah Al An'am [6]: 151). Furthermore, when the children are pious and grow up worshipping Allah, the gifts to their parents increase, making life more blessed. The parents' hard work in educating their children to become pious servants of God is the reason for the more blessings. Since parents inform their children, they have the fear of Allah. Furthermore, Allah says about the fruit of piety, "Whoever fears Allah, He will provide for him a way out and give him sustenance from a direction he did not expect.". (Surat Ath Thalaq [65]: 2–3)

The statements by P1, P8, and P12 indicate that the participants agree with the concept of "more children, many fortunes" as a realization of their belief in God. Every fetus that grows in the mother's womb has the right to life given by God. Religion for the Javanese religious groups or the strong Islamic upholders is a guide and a way of life (Hermawan et al. 2018). Therefore, the community's perspective on the number of children is influenced by the paradigm of thinking that human beings are creatures created by an almighty God, thus prohibited from performing actions that God does not like. This condition supports a previous study that the thinking paradigm affects the behavior of a person-specific community group (Brevers et al. 2021; Reling et al. 2018; Zhang et al. 2022).

*4.2. Every Child Has Their Fortune*

The participants stated that, as God's creatures, all humans are prohibited from killing children because they are afraid of poverty. Humans allow children to be born naturally in terms of birth spacing and number because each child brings his fortune. P3 stated the following.

All children born are guaranteed fortune by God. As a parent, I keep trying and working to meet all the needs of the children. We feel that having many children is not a problem. The more children, the more fortune because it makes me work diligently to meet the children's needs. (P3, lines 67–71)

P3's statement illustrates that the culture of "more children, many fortunes" developing in the community means each child has a fortune. This signifies that the culture of "more children, many fortune" relates to the values of children adopted by parents. Children are considered financial and psychological helpers when their parents are old. Furthermore, they help the family business and other siblings and continue the family

lineage. There is an implied meaning in P3's statement that parents with many children should not be afraid to be poor.

P8 also stated as follows:

> Many children have much luck, which is a belief in my family and me. Fortune has already been arranged. The fortune could be abundant wealth, health, and happiness that comes with every child. Allah SWT guarantees and provides it. (P8, lines 63–66)

P13 stated as follows:

> Yes, I have five children in school. Some are in elementary and middle school, and some are in college—the youngest is in 1st grade. I have many children because they are a gift from God. Suppose I had never thought that a child was a burden. The child is a gift from God. God promises that every creature would get a fortune. Therefore, I am sure fortune has been arranged. I believe that each child brings their own luck and fortunes. Why should it be limited? (P13, lines 71–76)

Another participant (P14) stated the same thing about the principle of life that everyone has their respective shares of fortune:

> The birth of a child in a family brings fortune. Something determined by Allah is the best, whether there are many children or no children. I am very grateful because we are blessed with three cute children. The analogy is straightforward. The Qur'an states that even reptiles have their fortunes guaranteed by Allah. Humans also have guaranteed fortunes. (P14, lines 63–67)

All participants are devout to their religion and believe that God the creator guarantees every living creature fortune. The parents, especially the fathers, are increasingly eager to earn a living by having many children. This finding shows that belief in God triggers intrinsic motivation to achieve optimal life goals (Exline et al. 2021; Thomas 2018).

### 4.3. Children as Family's Success Symbol

A child is significantly valued in a Javanese family because society places parents as the highest authority. In this case, they must respect, obey, and love their parents as an indication of good character in Javanese culture. Children and youth with good character and a polite attitude towards others are considered a success measured in terms of material wealth. Furthermore, the children are a success when raised as ideal people in the community, often referred to as *priyayi*.

In Javanese society, priyayi denotes a social class of people with respectable social positions, such as civil servants. Parents with financial ability tend to have many children and send them to a high level. The parents make their children priyayi because they are considered family treasures. Javanese parents strive to turn their children into priyayi. However, this requires obedience of the children to develop themselves according to the corridors set by their parents. In this case, parents see themselves as higher and must be obeyed by their children (Hasanah et al. 2019). One participant stated as follows:

> We view our children as a gift from God, which is very valuable. Therefore, we educate and take care of them with love. I have two sons and three daughters, all in school. When they grow up, they can get a good job and become priyayi (P10, lines 78–80)

In the past, Javanese women and men had different roles and were often treated differently. Boys were given more freedom than girls, especially regarding marriage. Girls were considered to have started their teenage years when they entered their menstrual period, while the sign for boys was to be circumcised. Although some Javanese children were circumcised at 8, the practice is typically carried out when the child is around 10 to 14 years old. In the past, menstruating girls would be immediately married to avoid

bad things. The marriage of teenage girls could be postponed to continue their education (Koentjaraningrat 1962). Differences in the views of parents towards children and their treatment based on gender still exist today, though not as extreme as in the past. Therefore, some modern Javanese families want a complete composition of children, including sons and daughters. They allow children to be born naturally until the composition is balanced. One participant (P5) stated as follows:

> PARTICIPANT (P5): I have 5 children, all girls. I still intend to get pregnant again because I hope to have a boy.
>
> INTERVIEWER: Why do you need a son, Ma'am? Aren't girls and boys the same?
>
> PARTICIPANT (P5): Yes. People say it is the same for boys or girls, but we long for a boy in our family to be complete. I have a pregnancy program to get a boy. Moreover, my husband wants a son. He said that when he was old, the son could help replace his father's role in society. (P5, lines 92–93)

In Javanese families, children assume the responsibilities of the parents when the parents grow unproductive, are weak, and need special attention. Children must be able to give sincere love and bear the financial burden of their unproductive parents.

*4.4. Children as Eternal Binders in Domestic Relationships*

In Javanese culture, children have a vital role in building a harmonious relationship between husband and wife (Aryanti 2017). They are happy gifts who strengthen the family bond (Zainuddin 2005). Moreover, children could be a reason couples remain loyal to each other. Such conditions seem to still exist in the life of modern Javanese society, as stated by a participant:

> About seven years ago, I almost divorced my husband because I still had no children after so many years of marriage. We often fought and blamed each other, it got worse when there was a third-party intervention. However, I finally got pregnant with my first child. A year later, I got pregnant again, and now I have three children, and my family is improving. When my husband and I have a problem, we immediately forgive each other. I feel sorry for the children when their parents fight. (P7, lines 82–87)

P9 also stated as follows:

> My wife and I almost divorced because we had no children after ten years of marriage. Finally, our extended family allowed us to adopt a child to keep our household harmonious. Ultimately, we can live together and be happy, though the child is not our blood. We love our little daughter with all our hearts. Five years after adopting a child, my wife became pregnant with twins and another baby two years later. When my wife becomes pregnant again in the future, we would never refuse it because children are the source of happiness in our family. We love all our children (P9, lines 90–96)

## 5. Conclusions

This study found that the philosophy of "more children, many fortunes" still exists in modern Javanese society, with many couples having three or more children. However, there seems to be a shift in the meaning and attitudes of parents toward children in extended families. When the philosophy first emerged, children were assumed to play a role in supporting the family's economic life. Many children became workers to help their parents in the garden or the fields; most Javanese people had a livelihood in farming and needed cheap and plentiful labor, though this condition has changed. However, modern Javanese parents still have many children in their families and interpret the belief that children equate to fortune differently from the past. Parents assume Allah will provide fortune for children in the family, regardless of their number. Therefore, children are no longer a source of labor for the family but are to be educated formally and informally. Because parents are

obligated to fulfill children's rights since they are gifts from the Creator. Following these results, future studies could determine the proportion of Javanese families with three or more children.

**Funding:** This research received no external funding.

**Institutional Review Board Statement:** The study was conducted in accordance with the Declaration of Helsinki, and approved by the Institutional Review Board of Komite Etik Penelitian Universitas Ahmad Dahlan (No. 012208106, 18 August 2022).

**Informed Consent Statement:** Informed consent was obtained from all subjects involved in the study.

**Data Availability Statement:** The study did not report any data.

**Conflicts of Interest:** The author declares no conflict of interest.

## Notes

1  Safii, "Konsep Kesempurnaan Hidup Orang Jawa: Sebuah Tinjauan Filologi Terhadap Serat Madurasa."
2  World Medical Association Declaration of Helsinki: Ethical Principles for Medical Research Involving Human Subjects.

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
