# Peer review of "Java Community Philosophy: More Children, Many Fortunes"

_genealogy, doi:10.3390/genealogy7010003_

Round 1
Reviewer 1 Report
I think this paper is making an interesting point about continuing cultural influences on fertility, using at the motivations of high fertility couples in Java, Indonesia. However, we need to know how stable these preferences are, for which we need to know what proportion of couples still have three or more children. I am sure these preferences and motivations have been changing as they have been in most parts of the world. I therefore recommend a table giving total fertility rates in Java over time, as well as the proportion of families with one, two and three plus children currently.
In addition to this, I have made several comments in the PDF of the paper and am attaching this marker up paper.
The language will need extensive editing, I hope you have a copy editor who can do this.

Author Response
Thank you for the review. Here I attach the results of my revision. I hope it's what you expected, thank you

Reviewer 2 Report
The topic of the article is interesting both with reference to the specific phenomenon invesatigated by the author and from a wider point of view.
Nevertheless two main points must be improved:
1. The logic structure of the article must be more clearly presented: the author should more explicitly present both question and hypothesis either at the end of section 2 or at the beginning of section 3. Similarly, the author should more explicitly discuss results with reference to these question and hypothesis in Conclusion
2. More information about method must be provided, in particular about concrete sampling strategies, interview schedule, strategies in the analysis of interviews.
Finally, since the article refers to Java community in the title whereas all the data are from Yogyakarta, the author should discuss this misalignment either in the section about method or in conclusion, reflecting upon potential distortion emerging from focussing only on this city.
Author Response
Thank you for the review. Here I attach the results of my revision. Thank you
